# EEG Fractal Analysis Reflects Brain Impairment after Stroke

**DOI:** 10.3390/e23050592

**Published:** 2021-05-11

**Authors:** Maria Rubega, Emanuela Formaggio, Franco Molteni, Eleonora Guanziroli, Roberto Di Marco, Claudio Baracchini, Mario Ermani, Nick S. Ward, Stefano Masiero, Alessandra Del Felice

**Affiliations:** 1Department of Neuroscience, Section of Rehabilitation, University of Padova, Via Giustiniani 3, 35128 Padova, PD, Italy; emanuela.formaggio@unipd.it (E.F.); roberto.dimarco@unipd.it (R.D.M.); stef.masiero@unipd.it (S.M.); alessandra.delfelice@unipd.it (A.D.F.); 2Villa Beretta Rehabilitation Center, Valduce Hospital, Via N. Sauro 17, 23845 Costa Masnaga, LC, Italy; franco56.molteni@gmail.com (F.M.); eleonora.guanziroli@gmail.com (E.G.); 3Stroke Unit and Neurosonology Laboratory, Padova University Hospital, Via Giustiniani 3, 35128 Padova, PD, Italy; claudiobaracchini@gmail.com (C.B.); mario.ermani@unipd.it (M.E.); 4Department of Clinical and Movement Neuroscience, UCL Queen Square Institute of Neurology, 33 Queen Square, London WC1N 3BG, UK; n.ward@ucl.ac.uk; 5Padova Neuroscience Center, University of Padova, Via Orus, 35128 Padova, PD, Italy

**Keywords:** neurophysiology, stroke, EEG, neuroplasticity, fractal analysis

## Abstract

Stroke is the commonest cause of disability. Novel treatments require an improved understanding of the underlying mechanisms of recovery. Fractal approaches have demonstrated that a single metric can describe the complexity of seemingly random fluctuations of physiological signals. We hypothesize that fractal algorithms applied to electroencephalographic (EEG) signals may track brain impairment after stroke. Sixteen stroke survivors were studied in the hyperacute (<48 h) and in the acute phase (∼1 week after stroke), and 35 stroke survivors during the early subacute phase (from 8 days to 32 days and after ∼2 months after stroke): We compared resting-state EEG fractal changes using fractal measures (i.e., Higuchi Index, Tortuosity) with 11 healthy controls. Both Higuchi index and Tortuosity values were significantly lower after a stroke throughout the acute and early subacute stage compared to healthy subjects, reflecting a brain activity which is significantly less complex. These indices may be promising metrics to track behavioral changes in the very early stage after stroke. Our findings might contribute to the neurorehabilitation quest in identifying reliable biomarkers for a better tailoring of rehabilitation pathways.

## 1. Introduction

Stroke is one of the most common brain disorders worldwide, affecting 17 million people each year. It is the second most common cause of death and physical disability [1]. In Europe, an ageing population and the escalation of risk factors will lead to an estimated 32% increase in DALYs (disability-adjusted life years) lost from 2015 to 2035 (from 609,721 to 861,878) [2] and related costs estimated at 15.9 billion [1]. Finding ways to improve outcomes is an urgent clinical and scientific priority, but this requires a better understanding of the underlying biological mechanisms of recovery in humans. This endeavour would be greatly aided by appropriate biomarkers to understand the biological phenotype differences between stroke patients [3].

Electroencephalography (EEG) is a non-invasive imaging technique that allows mapping electrical activity in the cortex. EEG is strongly influenced by the ongoing neurochemical processes that take place after a stroke. Several linear EEG indices have been suggested as markers of brain dysfunction after a stroke [4], e.g., EEG topographical distribution, power spectra and laterality coefficients [5,6], but the nonlinear dynamic properties characterizing the complex behaviour of the brain have not been fully elucidated in stroke survivors [7]. Indeed, EEG stroke induced modifications have been studied with linear analysis methods [8,9], such as spectral analysis or EEG topography, but they have never gained momentum and have not been incorporated into clinical practice. Their relative inadequacy is likely related to the intrinsic properties of the EEG signal: the electrical potentials recorded at scalp level are the coalescence of overlapping brain cell action potentials in time and space. Moreover, spectral analysis reduced the EEG power spectral quantification to specific frequency range of interest to establish a priori. Thus, nonlinear methods may reveal a more accurate description of cerebral electrical activity and an useful tool in understanding the mechanisms of neuronal plasticity after injury and during rehabilitation [7,10,11]. For instance, nonlinear dynamics analysis showed significant differences between lesion regions and normal regions in stroke survivors [12]. Lower values of entropy in post-stroke survivors compared to healthy adults in the fronto-central regions suggest that motor dysfunction is characterized by a loss of complexity of brain activity [10].

Signal complexity can be assessed using various nonlinear indicators. Among these, entropy-based algorithms are very popular and powerful tools for analyzing EEG [13,14,15], but their high computational cost may hamper their use, in particular in real-time applications and in clinical settings. Higuchi’s measure of Fractal Dimension (FD) and FD Residuals and Tortuosity [16,17,18] have instead a lower computational cost than entropy based-algorithms such as Sample Entropy (O(*n*) versus O($n^{2}$)). FD was introduced to estimate the features of irregularly shaped objects in which a similar pattern repeats itself at different scales and is calculated directly in the time domain [19]. Applied to biological systems, FD appears to be reduced in altered states of consciousness [17,18,20] and increases during cognitively demanding tasks [21]. On the other hand, other low computational cost nonlinear approaches, such as the Lempel-Ziv method [22], may overlook some information related to changes in the brain function because they are based on signal quantization, i.e., the EEG signal is reduced to a binary sequence (0–1).

In our previous works [17,18], we applied both entropy-based methods, such as Sample Entropy (SampEn), empirical permutation entropy (ePE) and empirical conditional entropy (eCE), and FD measures to EEG data in type 1 diabetic subjects. In particular, we verified if nonlinear EEG indices can be used to detect hypoglycemia with good accuracy: we trained and tested a neural network on EEG data to evaluate both individual measures and different combinations of complexity indices. Considering the indices individually, the best classification results were obtained by the Higuchi index. However, satisfactory results were obtained only when considering the fractal indices (i.e., Higuchi, Tortuosity and Residuals) together. The results slightly improved by adding ePE. Both fractal-dimension measures and entropy-based approaches suggest that the EEG signal, for hypoglycemia, describes a process that is more regular and less complex.

Authors in [23] computed FD on EEG signals with different algorithms (Higuchi, Katz and Multi-resolution Count Boxes) and proved that Higuchi index had the smallest variance for evaluating the EEG signal complexity in the same experimental condition compared to the other indices, being the most reliable to estimate the FD.

For the above reasons, we computed FD Higuchi and Tortuosity in this work.

In this study, we took into account that: (1) Clinical scales (e.g., Fugl-Meyer Assessment) are inadequate to discriminate between compensation and behavioural restitution [3]; (2) Healthy controls exhibit higher values of FD compared to stroke survivors with reduced functional/motor abilities [7,11]; and (3) Higher FD values are related to more complex brain activity [21]. In light of these considerations, we formulated the following research questions:

(Q1) What is the time course of the fractal dimension EEG indexes (FD Higuchi, FD Tortuosity) in the acute and early subacute stages post stroke?

(Q2) What is the topographical distribution of the FD measures in the acute and sub-acute post-stroke compared to healthy controls?

We hypothesized that FD indexes would gradually tend toward values similar to those of age-matched healthy control, with a progressive course over time.

## 2. Materials and Methods

We referred to the following time-line: ∼[0–48] h defined as hyper-acute phase, ∼[2–7] days as the acute phase, ∼[8–90] days as early subacute phase [3].

We collected resting EEG data of 51 post-stroke survivors in two different hospitals:

Dataset 1 of 16 participants recorded and clinically monitored in the hyperacute (<48 h) and in the acute phase (∼1st week after stroke) at the Stroke Unit of Padova Teaching Hospital;

Dataset 2 of 35 different participants recorded and clinically monitored during the early subacute phase from 8 days to 32 days after stroke and ∼2 months after stroke at the Neurorehabilitation Unit of Villa Beretta Valduce Hospital.

We recorded EEG resting-state in 11 age-matched healthy controls (Dataset 3).

To answer the research questions (i.e., Q1, Q2), we evaluated EEG FD measures at: T0, i.e., 30±21 h, and at T1, i.e., 7±2 days after the event for Dataset 1; $\tilde{T1}$, i.e., 15±7 days, and at T2, i.e., 54±9 days after the event for Dataset 2. We refer the reader to Figure 1 for the distribution of timepoints of EEG recording and clinical evaluation among participants in the two datasets. (In the caption of Figure 1, the shape α and scale β parameters of each Gamma distribution are reported. To compute the mean and the standard deviation of the normal distribution that these Gamma distributions approximate, we computed the mean as α∗β and the standard deviation as $\sqrt{\alpha *\beta^{2}}$.)

### 2.1. Participants

Inclusion criteria for stroke survivors were: first ever symptomatic stroke in the middle cerebral artery distribution [Partial Anterior Cerebral Artery Infarct (PACI) or Total Anterior Cerebral Artery Infarct (TACI)] [24], regardless of thrombolytic or interventional treatment, with no concomitant condition impeding motility—e.g., severe orthopedic or rheumatologic disease—without global or comprehension aphasia, right handed according to the Edinburgh Handedness Questionnaire [25] and able to provide informed consent.

#### 2.1.1. Acute Phase

Dataset 1 was collected at the Stroke Unit of the University Hospital of Padova at two timepoints: T0 and T1 between March 2017 and July 2018. Participants were 16 right-handed subjects: Six with right hemisphere lesion (age in the interval 29–82 years) and 10 with left hemisphere lesion (55–81 years). The caring physician of the Stroke Unit alerted the research team whenever a potential participant was admitted; one physician of the research team screened the subject and in case enrolled him/her. The same researcher also collected clinical and behavioral data; EEG was collected by two dedicated technicians at the same time of clinical-behavioral assessment.

#### 2.1.2. Early Subacute Phase

Dataset 2 was collected at the Villa Beretta Hospital at two timepoints: $\tilde{T1}$, i.e., at admission to the Neurorehabilitation Unit, and T2, i.e., at discharge, between January 2018 and February 2020. Participants were 35 right-handed subjects: 18 with right hemisphere lesion (38–79 years), and 17 with left hemisphere lesion (43–81 years). The physicians of the Rehabilitation Unit screened the subject, checked inclusion and exclusion criteria and enrolled him/her. The physician also collected clinical data; EEG was collected by a dedicated researcher at the same time of clinical assessment.

#### 2.1.3. Healthy Controls

Dataset 3 was collected at the Padova Neuroscience Center (PNC). Healthy right-handed healthy adults (31–76 years) were included in this study. Exclusion criteria were: (1) Already diagnosed neuropathy or sensation of tingling in the legs or feeling of having reduced sensitivity to feet or legs; (2) Diabetes mellitus; (3) Any rheumatological pathology; (4) Orthopedic problems (e.g., arthrosis); (5) History of fractures of the lower limbs or feet; (6) History of spinal surgery; (7) History of hip and/or knee prosthesis operations; (8) History of stroke even if completely recovered. Hypertension was not considered a contraindication if under drug treatment and with good control.

This study was approved by the ethical committee (protocol n. 444/2019, 18/06/2019) and all subjects gave their informed consent for inclusion before they participated in the study.

For further details on Dataset 1, 2 and 3, see Appendix C.

### 2.2. Experimental Protocol

#### 2.2.1. Acute Phase—Dataset 1

30-channel scalp EEG recordings (BrainAmp 32MRplus, BrainProducts GmbH, Munich, Germany) were acquired at T0 (30 ± 21 h) and T1 (7 ± 2 days) at a sampling rate of 5 kHz. The reference was positioned between Fz/Cz and ground anterior to Fz according to the 10/10 system. EEG data were acquired using an analogic anti-aliasing band pass-filter at 0.1–1000 Hz (as physically implemented in the amplifier). EEG data were recorded during 5-min resting state with open eyes.

During the same session of EEG recording, the clinical NIH Stroke Scale (NIHSS) [26] and Fugl-Meyer Assessment scale (FMA) [27] were also collected. The NIHSS is a 15-item neurologic examination stroke scale used to evaluate the signs of acute cerebral infarction on the levels of consciousness, language, neglect, visual-field loss, extraocular movement, motor strength, ataxia, dysarthria, and sensory loss (ratings for each item are scored with 3 to 5 grades with 0 as normal). The FMA is an index to assess the sensorimotor impairment in stroke recovery and rehabilitation trials (FMA maximum score which corresponds to full sensory-motor recovery for the upper extremity is 66 points and for the lower extremity is 34 points).

#### 2.2.2. Early Subacute Phase—Dataset 2

60-channel scalp EEG recordings (Compumedics Neuroscan, Compumedics, NC, USA) were collected at $\tilde{T1}$ (15 ± 7 days) and T2 (54 ± 9 days). EEG data were acquired at a sampling rate of 1000 Hz during 10-min resting state with open eyes, with the reference between Fz and Cz and the ground anterior to Fz positioned according to a 10/10 system, band pass-filtered at 0.1–1000 Hz and digitized.

During the same session of EEG recording, the clinical Action Research Arm Test (ARAT), Box and Block Test (BBT) and Nine-Hole Peg Test (N-HPT) were also collected. The ARAT is a 19 item observational measure used to assess upper extremity performance, coordination, dexterity and functioning with task performance rated—for each item—on a 4-point scale, ranging from 0 (no movement) to 3 (movement performed normally). The BBT measures unilateral gross manual dexterity. It is composed of a wooden box divided in two compartments by a partition and 150 blocks; the test administration consists of asking the participant to move, one by one, the maximum number of blocks from one compartment of a box to another of equal size, within 60 s (the BBT is scored by counting the number of blocks carried over the partition from one compartment to the other during the one-minute trial period). The N-HPT is used to measure finger dexterity and it is administered by asking the participant to take the pegs from a container, one by one, and place them into the holes on the board, as quickly as possible (the N-HPT is scored by counting the time needed to complete the test).

#### 2.2.3. Healthy Controls—Dataset 3

High-density EEG recordings were acquired inside a dimly lit sound-attenuated and electrically shielded room with the Geodesic Sensor Net with 256 electrodes (Electrical Geodesic Inc., Eugene, OR, USA). The recordings were sampled at 500 Hz, referenced to Cz. 5-min EEG data were recorded during resting state with open eyes.

### 2.3. EEG Pre-Processing

The data were processed in Matlab R2019b (MathWorks, Natick, MA, USA) using personalized scripts based on EEGLAB toolbox (http://www.sccn.ucsd.edu/eeglab, accessed on 31 December 2020) [28] and on [17,18]. The EEG recordings were band-pass filtered from 1 to 30 Hz (the optimal Chebyshev finite impulse response filters were designed using Parks–McClellan algorithm, the order was customized to minimize the error in the pass and stop bands) to remove signal drift and 50 Hz noise and down-sampled at 500 Hz. Eyes movements and cardiac activity were removed using independent component analysis (FastICA algorithm) based on the waveform, topography and time course of the component, and data were re-referenced to the average reference. Forty-one 2-s EEG epochs—i.e., non-overlapping segments of 1000 samples—were extracted for each participant during resting-state.

### 2.4. EEG Fractal Dimension Measures

The nonlinear indices, i.e., FD Higuchi and Tortuosity (see [16,17,18] and Appendix A for further details in the FD measures computation), were computed for each of these 2-s EEG epochs (41 epochs per recording). FD Higuchi considers the linear region of the fractal curve computed from the input data (i.e., 41 EEG epochs of 2 s) whereas the oscillatory behavior of the nonlinear part of the fractal curve—whose features depend on the periodicity of the signal itself—is evaluated from FD Tortuosity (and Residuals in Appendix A). FD Higuchi increases as the signal irregularity increases, whereas Tortuosity increases as the signal periodicity increases.

### 2.5. Statistical Analysis

Considering that the lesion location may cause an asymmetric EEG topography, we a-priori divided our participants with right (non-dominant) and left (dominant) hemispheric lesions in performing the statistical comparisons with controls. Because FD measures were not normally distributed, non-parametric tests were applied. We performed two-sided Wilcoxon rank sum tests to compare all FD features among healthy controls (Dataset 3) and each time-points in Dataset 1 (T0 and T1) and Dataset 2 ($\tilde{T1}$ and T2), dividing stroke survivors with right (non-dominant) and left (dominant) hemispheric lesions. To correct for multiple comparison, Bonferroni correction was applied to avoid Type I errors (i.e., incorrectly rejecting the null hypotheses): we accepted a probability of 0.16% of making a Type I error for Dataset 1 and of 0.08% for Dataset 2. Dataset 3 was spatially downsampled to 30 channels to be compared to Dataset 1 and spatially downsampled to 60 channels to be compared to Dataset 2.

A paired sample two-sided Wilcoxon signed rank test was performed for identifying significant differences over time between the clinical scales (p<0.05 was considered statistically significant).

Steps of the data analysis, described in the previous paragraphs, is schematically reported in Figure 2.

In the Section Results, FD values and their statistics are reported through topographies, i.e., EEG electrodes are placed onto the head following a geometrical array of even-spaced points. The value which refers to each electrode is color-coded (i.e., blue and pale blue depict lower values, while yellow and red depict higher values). The spatial points lying between electrodes are calculated by interpolation (i.e., calculating intermediary values on the basis of the value of its neighbors), and thus a smooth gradation of colors is achieved. This approach provides a much more accurate and representative view of the location of altered plotted values on the scalp.

## 3. Results

As reported in details in the FD features algorithm description in Appendix A, FD Higuchi and Tortuosity are computed after estimating from each EEG scalp recording the “fractal” (i.e., log(L(k)) versus log(k)) curve. We reported in Figure 3 as a representative example the curve computed for EEG channel Fz in three representative subjects: one stroke survivor from Dataset 1 at T0 and T1 (see respectively lines green and violet), one stroke survivor from Dataset 2 at $\tilde{T1}$ and T2 (see respectively lines dark violet and yellow) and one control from Dataset 3 (see red line). We can note that the first linear behaviour that is characterizing the curves has a wider extension for stroke survivors compared to the control. Moreover, the red curve (control subject) changes its curvature a few times more than stroke survivors. Neglecting the y-axis intersection values, the stroke survivors curves reveal a really similar pattern.

In Figure 4, we reported the median topographies over participants for the FD Higuchi and Tortuosity in all Datasets at all time-points. From a qualitative point of view, the topography obtained for Dataset 3 (controls) has a similar distribution of the one obtained in healthy control subjects in a previous study [7,11].

Comparing the topographies in acute phase vs controls, we obtained statistical significant differences in all channels (*p_value_*
$<1.62\times 10^{-4}$). FD Higuchi was higher in controls (i.e., range [1.1098, 1.3447]) than in stroke survivors (i.e., range [1.0222, 1.0835]), revealing an EEG complexity loss in stroke. Moreover, FD Tortuosity has 3-time higher values in controls ([0.0767, 0.1659]) than stroke survivors ([0.0164, 0.0604]) suggesting the presence of many detectable oscillations in controls revealing a richer EEG signal.

In comparing the topographies in subacute phase with the control ones, we obtained statistical significant differences for EEG recording in all channels, reaching p-values lower than ones in acute phase. We can extend the same observations comparing controls with stroke survivors in acute phase. Most likely, the subacute phase cohort reveals a greater impairment compared to the acute phase one. Unfortunately, we cannot quantitatively test this hypothesis, being the clinical scales (see Table 1 and Table 2) collected in the two cohorts (i.e., Dataset 1 and 2) different. We can only note that the NIHSS improvement is statistically significant in the acute phase cohort (p=0.0015) whereas only one clinical scale (BBT) out of three is statistically significant in the subacute phase cohort with a *p* value (p=0.0388) near to the statistical threshold (p=0.05). Moreover, the high variability (i.e., standard deviation) of the clinical scales in Dataset 2 (Table 2) reveals that many stroke survivors were unable to successfully perform the tasks required by clinical testing (i.e., 10 participants for BBT score, 17 for N-HPT and 5 for ARAT).

## 4. Discussion

Our results demonstrate that nonlinear properties of the EEG signal reflect modifications of brain state after stroke. Higuchi values were significantly lower after a stroke throughout both the acute ([1.0224, 1.0718] at T0; [1.0222, 1.0723] at T1) and early subacute stage ([1.0310, 1.0723] at $\tilde{T1}$, [1.0317, 1.0835] at T2) compared to healthy subjects ([1.1098, 1.3447]), reflecting a brain activity which is significantly less complex. On the other hand, Tortuosity values revealed a brain activity more modulated in controls ([0.0767, 0.1659]) compared to stroke survivors in acute ([0.0171, 0.0517] at T0; [0.0164, 0.0537] at T1) and early subacute stage ([0.0228, 0.0604] at $\tilde{T1}$, [0.0222, 0.0602] at T2).

In the acute phase (1–7 days after stroke), FD measures remained strongly impaired compared to those of healthy controls, with a consistent reduction in complexity. Intra-subject comparison between admission and discharge from Stroke Unit show a non-significant increase of complexity. Indeed, only NIHSS modifications were significant (i.e., on average, 4.75 at T0 vs. 1.80 at T1), whereas FMA modifications were not (i.e., on average, 55 at T0 vs. 60 at T1), most likely due to the ceiling effect of FMA scale already described in studies with a similar design [29]. Our participants reached almost normal values of both NIHSS and FMA, except for a few severely impaired participants.

As recently described for oscillatory brain activity (e.g., power spectra), EEG modifications in the acute stage show a different time course from neurobehavioural changes as detected by the NIHSS and FMA scores. The authors describe an improvement over 6 months of EEG linear parameters, while clinical score did not substantially differ. It is likely that both scales are unable to detect more nuanced and subtle changes in behaviour due to the same ceiling effect we reported for FMA.

In the cohort recorded in the subacute stage, FD measures remained strongly impaired, with a consistent reduction in complexity. No significant behavioural improvement occurred except for BBT. While other studies [29] did in fact observe EEG changes over 6 months in a sample of post-stroke persons discharged at home, our sample comprised severely affected persons admitted to a Rehabilitation Unit. The follow up was limited at two months after stroke. This time-frame may be insufficient to detect changes both at a behavioral and EEG level, although the severity of the clinical picture may have played a consistent role in hampering recovery. As for the research question of the present study, we have still no clear-cut criteria for clinical trials on stroke recovery to adequately select participants. Conversely, we cannot also rule out an insufficient sensitivity of FD to monitor changes in this group.

The quest for non-invasive biomarkers of cerebral impairment and recovery led to the development of quantitative EEG analysis based on linear properties (e.g., power spectra and EEG topographies). These methods, although of interest from a research perspective, have only partially been translated into clinical practice and often not for stroke but, for example, as monitoring during anesthesia. A recent work addressing the question whether EEG spectra and Bispectral Index correlate with clinical recovery in stroke during the first six months found a modification of EEG index not related to behavioural (NIHSS, FMA) changes [29]. Many factors may have contributed to EEG linear measures scarce uptake: a potential obstacle may have been the expertise in neurophysiological signal interpretation as well as their relatively low prognostic value. Another obstacle may have been that the application of more advanced EEG signal analysis techniques is often perceived by the clinician of the acute care setting as alien to their expertise.

Non-linear EEG signal measures may thus embody a double advantage: they reflect more accurately brain processes and provide a single metric, which may be more intuitive for the clinician. In the last decades, the application of nonlinear dynamics to EEG opened up a range of new perspectives for the investigation of brain function and is developing toward a new interdisciplinary field of nonlinear-brain-dynamics [30]. Nonlinear measures allow extraction of information complementary to the classical linear features, e.g., power spectrum, to enable a comprehensive characterization of the EEG time-series. Among nonlinear techniques, FD measures, besides having no need for storage memory, are classified as linear-time algorithms—i.e., they have a time complex O(kN) where N is the number of samples and k the FD scale [18], making FD measures exploitable in real-time applications. Nonlinear methods have been applied so far to stroke only in anecdotal studies [7,11], using NIHSS as clinical assessment and a 19-channel EEG recording system. The authors describe an increase of EEG complexity over time not coupled by significant NIHSS changes.

Nonlinear dynamics measures find potential applications to Disorders of Consciousness, supporting the challenge of the differential diagnosis between states [31], but FD have not yet been deployed in this field.

## 5. Conclusions

FD is a promising and sensible analysis method, able to detect even subclinical modifications after a stroke. As our results demonstrate, it provides an insight into the neuronal reorganization process after a stroke, which appears to be not always paralleled by EEG changes, as in fact linear measures suggest. This higher sensibility may pave the way for a more comprehensive understanding approach after cerebral lesions, offering a diverse perspective with the potential for becoming a screening and monitoring tool of neuronal restoration.

## 6. Limitations

The main limitation of this study is the use of different clinical tests in the two cohorts. A longitudinal study, which was in the original design of the Stroke Unit cohort, was not completed, thus the integration for the subacute phase with an additional cohort.

## Figures and Tables

**Figure 1 entropy-23-00592-f001:**
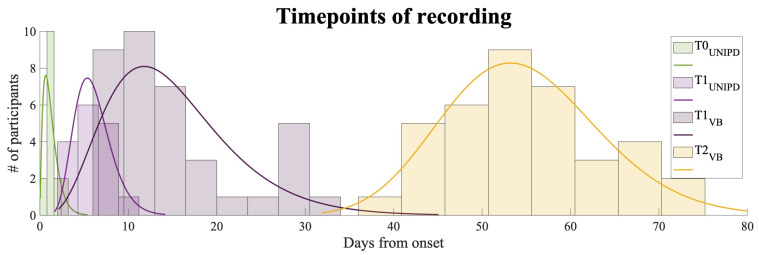
Histograms reporting the timing of EEG recording and clinical and behavioral data collection for the two datasets recorded in Padova (PD) and in Villa Beretta (VB). On the *x*-axis the days after the event in which the EEG recording and clinical evaluation were performed and on the *y*-axis the number of participants that were evaluated at that day. The distribution of the participants evaluated at T0 at the Stroke Unit of Padova is represented in green; the distribution of the same participants evaluated at T1 is represented in violet; the distribution of the participants evaluated at $\tilde{T1}$ at the Neurorehabilitation Unit of Villa Beretta is represented in violet too and the distribution of the same participants evaluated at T2 is represented in yellow. A Gamma distribution *gamma*(*α*, *β*) was fitted to each subset of data: Hyper-acute phase represented in green was quantified as T0_*UNIPD*_∼*gamma*(2.13, 0.59); Acute phase in violet as T1_*UNIPD*_∼*gamma*(9.58, 0.7), T1_*VB*_∼*gamma*(4.57, 3.29); Early subacute phase in yellow as T2_*VB*_∼*gamma*(38.28, 1.42).

**Figure 2 entropy-23-00592-f002:**
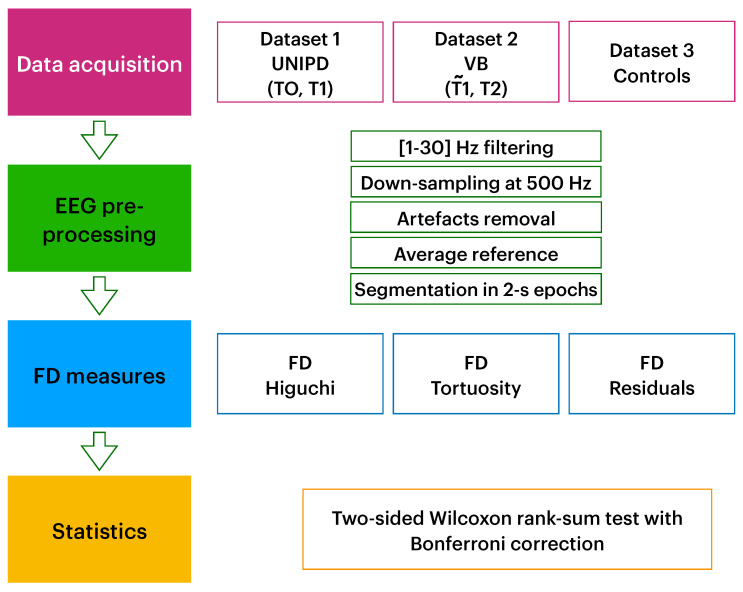
Schema of data analysis.

**Figure 3 entropy-23-00592-f003:**
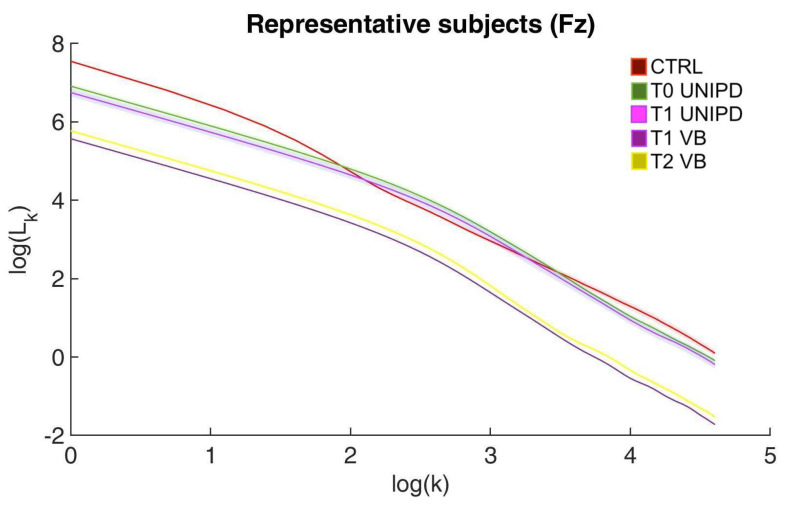
Fractal (i.e., log(L(k)) versus log(k)) curve computed for three representative subjects at EEG channel location Fz. The red line represents the median of the curves computed from the 41 EEG epochs of 2 s in one healthy control. The green and violet lines stand for the median of the curves computed from the 41 EEG epochs of 2 s at T0 (green) and at T1 (violet) for one stroke survivor of Dataset 1. The dark violet and yellow lines stand for the median of the curves computed from the 41 EEG epochs of 2 s at $\tilde{T1}$ (dark violet) and at T2 (yellow) for one stroke survivor of Dataset 2. Shaded areas represent ± standard deviation for each curve.

**Figure 4 entropy-23-00592-f004:**
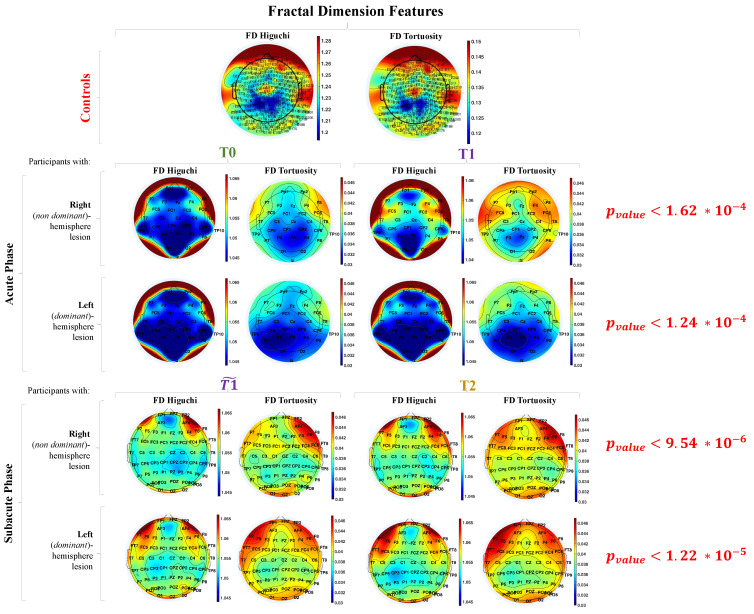
Topographic maps of the median values of FD Higuchi and Tortuosity during resting-state in healthy controls (first row) and during acute (second and third row) and sub-acute (fourth and fifth row) phase in stroke survivors. Second and fourth rows refer to stroke survivors with right-hemisphere lesion, third and fifth rows to stroke survivors with left-hemisphere lesion.

**Table 1 entropy-23-00592-t001:** Acute phase—Dataset 1: NIHSS and FMA.

Clinical Scale ($p_{\mathrm{value}T0\mathrm{vs}.T1}$)	T0	T1
NIHSS (*p* = 0.0015)	4.75 ± 2.23	1.80 ± 2.20
FMA (*p* = 0.1066)	55.00 ± 17.25	60.00 ± 16.36

**Table 2 entropy-23-00592-t002:** Subacute phase—Dataset 2: BBT, N-HPT and ARAT.

Clinical Scale ($p_{\mathrm{value}\tilde{T1}\mathrm{vs}.T2}$)	$\tilde{T1}$	T2
BBT (*p* = 0.0388)	16.67 ± 16.99	26.31 ± 21.20
N-HPT (*p* = 0.557)	22.26 ± 27.51	18.25 ± 20.97
ARAT (*p* = 0.0706)	29.41 ± 26.14	36.94 ± 23.79

## Data Availability

The data presented in this study are available on request from the corresponding author. The data are not publicly available due to their containing information that could compromise the privacy of research participants.

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
