# Peer review of "EEG Fractal Analysis Reflects Brain Impairment after Stroke"

_entropy, 2021, doi:10.3390/e23050592_

Round 1

Reviewer 1 Report

This is indeed an interesting approach to analyse brain impairment after stroke. The statistical analysis could be improved, and the color maps could be better explained in the captions.

The measure of fractality can be appropriate but other entropic approaches can certainly be considered, for example, based on Permutation Entropy and related modifications. You could mention those kind of approaches. I suggest the following reference:

Mammone N, Ieracitano C, Adeli H, Bramanti A, Morabito F.C., Permutation Jaccard Distance-Based Hierarchical Clustering to Estimate EEG Network Density Modifications in MCI Subjects, IEEE Transactions on neural networks and learning systems", 29, 2018, pp. 5122-5135, ISSN: 2162-237X.

Author Response

Please, see attachment

Reviewer 2 Report

This work presents the fractal-based analysis of EEG signals to study the brain impairment after stroke, where the Higuchi index and Tortuosity values are considered as fractal measures. In general, the topic is interesting and the manuscript is well-written; yet, some issues have to be modified in order to clarify the methodology and its applicability.

There are other fractal algorithms (e.g., Katz, Box dimension, etc.). It is not clear how and why the Authors chose Higuchi and Tortuosity values. Other fractal algorithms have to be considered since in this work a fractal study is presented.

The features of the datasets and participants are not clear. Please use tables to present such information. For participants, include sex, age, weight, and other relevant features. In general, the datasets used are small and limited in terms of diversity; in this regard, can your results be considered reliable?

It is not clear why the EEG data were filtered at 0.1-1000 Hz. Higher frequencies could provide useful information for your work.

A filtering from 1 to 30 Hz is also applied; then, a down-sampling at 500 Hz is carried out. These filtering stages are not clear nor justified.

Which is the impact of using different filters?

A flowchart for your processing methodology will be useful.

Why is the two-sided Wilcox signed rank test used? There are other methods such as ANOVA or Kruskal Wallis. Why is p<0.05 statistically significant?

For the results of tables, boxplots are recommended.

Numerical results and plots for the fractal algorithms during the different phases are required in order to see the evolution of such indices (maximum and minimum values, trends, etc.)

Please improve the quality of figures (e.g., font size)

Numerical conclusions are recommended.

Round 2

Reviewer 1 Report

I've found this paper of interest. It Is well written and organizer. The figures are truly complementare ti the text. I appeciate the quality of replies ti reviewers.

Reviewer 2 Report

All the comments and suggestions have been properly addressed. This Reviewer recommends the manuscript acceptance.